# Illegal and Exploitative Sand-Digging Activities Could Be Managed to Create Suitable Nesting Habitats for Blue-Tailed Bee-Eaters (*Merops philippinus*)

**DOI:** 10.3390/ani13061112

**Published:** 2023-03-21

**Authors:** Canchao Yang, Gang Lu, Ting Cai, Xiaogang Yao, Yan Cai

**Affiliations:** 1Ministry of Education Key Laboratory for Ecology of Tropical Islands, College of Life Sciences, Hainan Normal University, Haikou 571158, China; 2Haikou Duotan Wetland Institute, Haikou 570100, China; 3Kuankuoshui National Nature Reserve, Suiyang 563300, China

**Keywords:** avian reproduction, human disturbance, *Merops philippinus*, negative interference, positive interference

## Abstract

**Simple Summary:**

Commercial sand digging initially created suitable nesting habitats to attract blue-tailed bee-eaters but subsequently reduced their breeding success by destroying their nests. However, commercial sand digging can be modified to create suitable and stable nesting habitats for bee-eaters.

**Abstract:**

The development of human society often interferes with wild animals and their natural habitats. Interference during resource exploitation is mostly negative; however, in some cases, it can be positive and even have significance for some species. In this study, we investigated the number of blue-tailed bee-eaters (*Merops philippinus*), a species under ‘state protection category II’ in China, between controlled and manipulated nesting habitats from 2017 to 2022. Our results indicated that commercial sand-digging activities, either illegal or approved, initially created suitable nesting habitats to attract blue-tailed bee-eaters but subsequently led to damage of nests or nesting habitats. However, sand digging can be modified by avoiding the breeding season to provide safe and suitable nesting habitats for bee-eaters. The number of breeding birds more than tripled when digging during the breeding season was avoided. We also found that conventional conservation strategies, which strictly prohibited sand-digging activities, did not contribute to the nesting habitats of bee-eaters. This study enriches the theories of conservation biology and emphasizes the importance of dialectical thinking regarding exploitative and seemingly destructive activities.

## 1. Introduction

Human activities related to resource exploitation have a profound impact on the functions and services of numerous ecosystems on Earth, changing the living conditions and fitness of a large number of organisms [1]. Different forms of human activity cause different levels of disturbance, but many leads to negative interference with animals. For example, deforestation for agriculture or development projects destroys habitats for many plant species and animals that depend on those plants for food and shelter. Selective logging alters the vegetation structure and reduces animal species richness compared to that in unexploited forests [2]. Similarly, hunting has been identified as a major driver of biodiversity losses. In the tropics, the abundance of birds and mammals, respectively, was found to be 58% and 83% lower in hunting areas than that in non-hunting areas [3]. The increasing intensity of logging and grazing results in a significant loss of suitable habitats for different species, which could lead to changes in the competitive dynamics among them [4]. Furthermore, nature-based recreational activities have negative effects on bird diversity in a variety of habitats in different climatic zones and regions of the world [5]. The rapid increase in recreational activity during the summer has greatly reduced the number of shorebirds and gulls present in coastal areas, limiting the capacity of such areas to act as a post-breeding stopover location and significantly reducing the time available for native species to hunt prey [6]. In addition to these negative effects, human activities also bring ecological traps to reduce wildlife fitness. The concept of an ecological trap was originally formulated in studies with birds but has now been observed in various animal species worldwide [7]. It is a phenomenon in which animals are attracted to an environment that appears to be suitable for their survival, but in reality, it is not [8]. This can lead to a decline in population and even the extinction of the species.

Nevertheless, some exploitative activities by humans that have negative effects on some animals may have positive effects on others. For example, agricultural exploitation harms native forests and animal habitats, but the resulting open environment is favorable to the cowbird (*Molothrus* spp.), a brood parasite that has rapidly expanded its breeding habitats to utilize a variety of additional species that did not previously experience parasitism [9]. Because of the absence of coevolutionary interactions with cowbirds, these new hosts have been driven to the edge of extinction [10,11]. Thus, the exploitative activity of humans destroyed the native forest, which created suitable habitats for some species but had a negative impact on others. Theoretically, if an exploitative activity could benefit some species, it could be applied or manipulated in an appropriate context to contribute to biological conservation. However, such application studies are not common in conservation biology.

Bee-eaters (*Merops* spp.) are cavity-nesting birds that require suitable substrates, such as sandy banks, to excavate for nesting [12,13,14]. Such nesting habitats are easily reconstructed, unlike those of many forest-dwelling species [15]. Bee-eaters avoid nesting in sandy banks with dense vegetation and abandon colony sites that have become overgrown [16]. Therefore, removing vegetation from slopes can increase the number of breeding birds and improve their breeding habits [14]. We found that illegal and exploitative sand-digging activities can simultaneously create suitable nesting habitats and damage the nests of blue-tailed bee-eaters (*Merops philippinus*), which are ranked as a ‘category II’ state-protected species in China (http://www.forestry.gov.cn, accessed on 5 December 2022). We used controlled and manipulated studies from 2017 to 2022 to provide a case for application studies and indicate that illegal and exploitative activities by humans can inspire conservation management to create a suitable nesting habitat for a species of concern.

## 2. Materials and Methods

### 2.1. Study Site and Study Animals

This study was conducted in six nesting habitats of bee-eaters from March to September of 2017–2022 in Haikou City (19°32′ N–20°05′ N, 110°10′ E–110°41′ E), Hainan Province, China (Table 1). The six sites comprised Jinhui (JH), Jinshawanyi (JY), Jinshawansi (JS), Jinshawan Wetland Reserve (JWR), Yongqing (YQ), and Wuyuanhe Bee-eater Reserve (WBR). Each study site was used as either a negative control, positive control, or manipulated treatment group during the investigation (see Section 2.2 for more details). The studied species, the blue-tailed bee-eater, is a non-passerine bird in the family Meropidae. It is a summer breeder in most areas of southeastern Asia but has been identified as a resident of Hainan Island, China [17,18]. Bee-eaters are cavity-nesting birds that excavate caves for reproduction [16]. They were ranked as ‘state protection category II’ in 2021 because of their limited distribution and low-quality nesting habitats (http://www.forestry.gov.cn, accessed on 5 December 2022). Before this study, we found that sand-digging activities for real estate projects could create suitable sand walls in which bee-eaters could excavate breeding caves. Such sand-digging activities are conducted to supply sand for use in construction and are either illegal or approved by local governments. Before sand-digging activities, sand walls at the study sites were covered by vegetation, making them unsuitable for bee-eater nesting (Figure 1a). Sand digging results in bare and steeply vertical sand walls, which are suitable for bee-eater nesting (Figure 1b). However, irregular and unpredictable sand-digging activities during the breeding season commonly destroy nests, eggs, and/or chicks of bee-eaters. Furthermore, when the sand walls are abandoned for more than 1 year after sand digging, rainwater and wind erosion result in sand that is too loose for bee-eater excavation, transforming them into unsuitable nesting habitats (Figure 1c).

### 2.2. Field Procedure

Each of the six study sites was assigned to one of three groups: negative control, positive control, or manipulated treatment. JH, JY, and JS formed the negative control group, where illegal and exploitative sand digging occurred during 2017–2022. These three negative control sites covered a total area of 4.5 ha (Table 1); this area was demarcated as a site to investigate the number of breeding birds without any intervention from conservation or manipulation. JWR (4 ha) was classified as the positive control group, in which illegal sand digging was only identified in 2018, and strict protection from sand digging was enforced from 2019 to 2022. Therefore, the positive control group represented a scenario of conventional conservation management.

YQ and WBR formed the manipulated treatment groups. At YQ, sand digging was absent in 2017–2020 and then manipulated sand digging was conducted 1 month before the reproduction of bee-eaters (March) in 2021 and 2022. At the WBR site, sand digging was absent in 2017, but manipulated sand digging was conducted in March every year from 2018 to 2022 (Table 1). These two treatment groups covered a total area of 3 ha; this area was demarcated to test the effect of manipulated sand digging as a conservation management strategy that may provide suitable nesting habitats for bee-eaters. In the first year at WBR, manipulated sand digging comprised the use of excavating machinery to excavate the sand walls; this was conducted in a controlled manner to create a standard vertical slope. From the second year onward, volunteers were recruited to maintain the slope of the sand walls using handheld shovels. For YQ, all manipulated sand digging was conducted using shovels.

We investigated the number of bee-eaters at the six study sites twice a month, from March to October. We used binoculars Swarovski EL 10 × 32 WB to observe and count all individuals of bee-eaters in each study site. Because the bee-eaters become colonies around the sand walls during breeding season, it was feasible to count all individuals in the study sites. The numbers of bee-eaters at the six study sites were counted at the same time to avoid repeat counts between study sites. For each study site, the largest number that was counted during the course of the investigation was used to represent the number of birds. According to our investigations, the bee-eaters began visiting the sand walls in April and reached peak breeding numbers in May of the same year (Figure 2). Therefore, we used the number of birds in May of every year as a representation of the year. We also noted the status of nesting habitats (i.e., whether they were destroyed by the exploitative sand digging) during the investigation.

### 2.3. Statistical Analyses

The generalized linear mixed model based on Markov chain Monte Carlo techniques (MCMC-GLMM) was used to compare the number of birds among different treatment groups. In the model, the number of birds was the response variable, and the treatment (but not the group) and year were the fixed effects, while the colony (i.e., the six study sites) was the random effect. The treatment, rather than the group, was used as a fixed effect because some groups included data before and after manipulation. For example, the YQ colony had no sand-digging activities in 2017–2020 but received manipulated sand digging in 2021 and 2022 (Table 1). Therefore, the fixed effect of treatment was classified as no sand digging, exploitative sand digging, and manipulated sand digging. The interaction between treatment and year was also tested. Logistic regression with a Poisson function of error distribution was then used to investigate the changes in the number of birds over the years among different groups. Here the number of birds was the response variable, while the group (negative control, positive control, and manipulated treatment) and year were the fixed effects. The interaction between the group and year was also investigated. The MCMC-GLMM and logistic regression were run using the *MCMCglmm* and *robust* packages, while the figures were generated with the *ggplot2* package in R (version 4.1.0) for Windows (https://www.r-project.org, accessed on 21 December 2022).

## 3. Results

The dynamic changes in the number of bee-eaters of negative control groups exhibited an irregular fluctuation over the years (Figure 3) because the suitable nesting habitats were destroyed by irregular exploitative sand digging. Instances of exploitative sand digging occurred before and after nest building, in different breeding stages, and the range and intensity of sand digging also varied. Sand digging before the nest-building period created suitable nesting habitats for the bee-eaters, but one or more subsequent sand-digging events during the breeding period resulted in the partial or total destruction of the nests (and even the eggs or chicks), depending on the range and intensity of the sand-digging activities (Figure 1d). Such irregularly timed sand digging led to the irregular fluctuation of the number of birds over the years. However, the bee-eaters in the positive control group were only present in the year in which exploitative sand digging occurred and were absent in the following 4 years after sand digging was strictly prohibited (Figure 3). In the manipulated treatment groups (YQ and WBR), the number of bee-eaters persistently increased every year (Figure 3). At WBR, excavating machinery and shovels were used to create and maintain a suitable nesting habitat for the bee-eaters (Figure 4), and the number of birds more than tripled over the three years of our study.

The results of MCMC-GLMM and logistic regression were consistent with the observations. Both the treatment and year significantly predicted the number of birds (Treatment: *P*-MCMC = 0.004, Year: *P*-MCMC = 0.026, MCMC-GLMM; Table 2), indicating that the number of birds differed among different treatments and years. The interaction between the treatment and year was also significant (Table 2); thus, the number of birds in each year varied with different treatments. Further logistic regression analyses revealed that both the group and interaction between the group and year significantly predicted the number of birds (Group: *Z* = −5.36, *P* < 0.001, Group × Year: Z = 5.35, *P* < 0.001, Logistic Regression; Table 3). This implies that the year-to-year changes in the number of birds differed between different groups.

## 4. Discussion

### 4.1. Sand Digging as an Unpredictable Threat

Based on our results, the number of bee-eaters observed when illegal and exploitative sand-digging activities occurred fluctuated over the years and was lower than that when sand-digging was controlled. The illegal and exploitative sand-digging activities by humans initially created suitable nesting habitats for bee-eaters. However, the number of breeding bee-eaters in these circumstances fluctuated dramatically over the years because such sand-digging behavior was irregular and unpredictable. The exploitative sand-digging activities occurred at any time of the year, before or after the nest-building of the bee-eaters. Our results showed that the nests, eggs, and/or nestlings of the birds were destroyed by such irregular sand digging during the breeding season. Therefore, illegal and exploitative sand-digging activities before the breeding season initially produced suitable nesting habitats to attract the bee-eaters but subsequently led to the damage of their nesting habitats or nests before or after they bred. Such circumstances may be regarded as an ecological trap [8] because the nesting habitat of bee-eaters was degraded by exploitative sand-digging activities, but these activities before the breeding season acted as an original cue that misled the birds into behaving as though the sand walls were still a suitable habitat. Blue-tailed bee-eaters only have one breeding attempt per year. These intermittent sand-digging activities could be very costly for bee-eaters because sand-digging during their breeding season (especially during the chick stage) may destroy their reproductive output for that year.

### 4.2. Prohibiting Sand Digging as Conventional Conservation

Although illegal and exploitative sand-digging activities were harmful to the bee-eaters, strict prohibition of such activities did not create suitable nesting habitats. The result of the positive control indicated that after sand digging was prohibited, no bee-eaters chose to build nests on the sand walls because rainwater and wind erosion caused the sand walls to become too loose for nesting excavation (Figure 1). Furthermore, sand walls covered by vegetation were also devoid of any bee-eater nesting activities (Figure 1). According to our observations, bee-eaters prefer bare and steeply vertical sand walls for nesting, probably because they reduce the risk of nest predation, especially from snakes, as such sand walls are difficult for snakes to climb [19,20].

### 4.3. Manipulated Sand Digging as Conservation Replacement

Compared with illegal and exploitative sand-digging activities, controlled and well-timed sand-digging was able to provide safe and suitable nesting habitats for bee-eaters as a substitute for conventional conservation (Figure 4). Our results indicated that the number of bee-eaters on sand walls significantly increased after the sand walls received manipulated digging. Furthermore, the number of birds increased with years under manipulated treatment. This implies that manipulated sand digging was a feasible way of conservation management. However, this conservation management strategy would not have formed without the discovery of the impact of illegal and exploitative sand digging on the reproduction of bee-eaters. Therefore, exploitative sand digging played a role as a precondition that inspired the idea of manipulated sand digging. The results of our 6-year study showed that such manipulation of sand digging was feasible and sustainable for the conservation of nesting habitats for blue-tailed bee-eaters. This study provides applied research in conservation biology and demonstrates that illegal and exploitative activities by humans can inspire conservation management strategies to create a suitable nesting habitat for a bird species.

### 4.4. Conservation Management and Education

Biological conservation education should not be conducted in the manner of an armchair strategist, limited to classrooms and theories; education should be practical and experiential [21,22]. The development of birding (or bird-watching) in China provides an excellent example of this kind of education. Twenty years ago (in the early 2000s), conservation biology was a new branch of science that received insufficient attention from the public in China. Homiletic education on biodiversity and conservation was common at that time, and there was little communication or dialogue between conservation biologists and students or members of the general public. However, bird watching became popular, developing rapidly over the following years in major cities and, subsequently, sweeping across other cities of the nation [23]. Birding in cities provides an opportunity to enjoy the beauty of wild birds and also acts as an agent and medium for understanding nature and its value [24,25]. Because visiting natural forests and watching wild animals were not common activities for most citizens, birding was an important activity, and cities acted as a necessary place to connect people and nature [24,26]. Empathy is necessary and effective for wildlife conservation education [27]. The blue-tailed bee-eater in this study could act as a case of a flagship species to arouse public interest in wildlife conservation. Therefore, the manipulated sand-digging project in Haikou City (Figure 4) has provided an integration of both conservation and education, showing the public that not only natural environments but also the cities they reside in are accountable for providing suitable habitats for wild animals.

### 4.5. Perspective and Suggestion

Other research has implied that some exploitation activities can have two effects: intensive logging can alter vegetation structure and reduce the richness of animal species [2], but low and medium levels of logging could increase animal diversity [28,29]. Similarly, this study proved that exploitative sand digging had a negative impact on the reproduction of bee-eaters but that standardized and well-timed sand digging could provide suitable nesting habitats for them. Therefore, this study enriches theories of conservation biology and emphasizes the importance of dialectical thinking on exploitative and seemingly destructive human activities. Although the manipulated sand-digging project for bee-eaters has been chosen as one of the ‘100+ Biodiversity Positive Practices and Actions Around the World’ (https://news.cop15-china.com.cn, accessed on 5 December 2022), there are some additional suggestions that could improve conservation management in the future. First, illegal sand digging should be prohibited, while other commercial sand-digging activities approved by the local government should be modified to avoid damage to bee-eaters’ nests. Such modifications include creating loose sand walls, covering these with plastic film after digging is completed, and timing sand-digging activities to avoid the bee-eater breeding season. Considering that bee-eaters have been ranked as a ‘state protection category II’ species, it is possible and feasible to put forward such stipulations by law. Second, although the yearly manipulation of sand digging could provide sustainable nesting habitats for bee-eaters, the sand walls will become reduced over the years. Therefore, finding a replaceable material that simulates the characteristics of sand walls but is resistant to rainwater and wind erosion may increase the sustainability of nesting habitats for bee-eaters in the future. Additionally, although sand digging provided suitable nesting sites for the bee-eaters, further studies that consider other living conditions, such as foraging and roosting sites, are needed for a full conservation evaluation of this species in the future. Finally, we also emphasize that exploitation, especially on a large scale, still causes disastrous worldwide loss of biodiversity, and other unknown living organisms may be negatively influenced by sand-digging activities. Therefore, further studies are needed to investigate such potential effects.

## 5. Conclusions

Our study indicates that commercial sand-digging activities initially create suitable nesting habitats for bee-eaters but may subsequently reduce their breeding success. However, such sand digging can be modified and manipulated to produce safe and suitable nesting habitats for bee-eaters. Therefore, seemingly destructive exploitation activities can have two sides; dialectical thinking can aid in extracting beneficial characteristics, which may contribute to the management of biological conservation.

## Figures and Tables

**Figure 1 animals-13-01112-f001:**
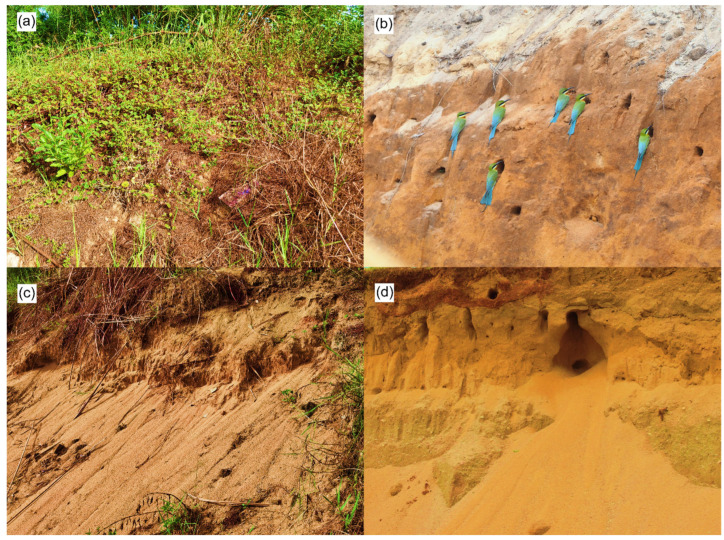
Different types of sand walls. (**a**) Before sand-digging activities, this sand wall was covered by vegetation, making it unsuitable for nesting by blue-tailed bee-eaters (*Merops philippinus*). (**b**) Sand digging resulted in a bare and vertical slope where bee-eaters could easily excavate breeding cavities. (**c**) After 1 or 2 years of abandonment and no sand digging, the sand wall became too loose for cave excavation due to rainwater and wind erosion. (**d**) A sand wall with bee-eater breeding cavities, destroyed by exploitative sand digging.

**Figure 2 animals-13-01112-f002:**
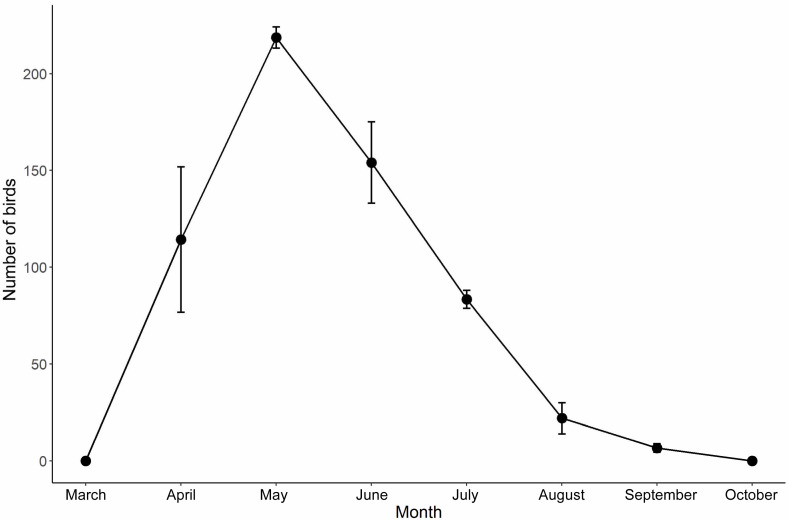
Number (mean ± stand error) of blue-tailed bee-eaters (*Merops philippinus*) that bred in sand walls of sites in Haikou city in 2019–2021.

**Figure 3 animals-13-01112-f003:**
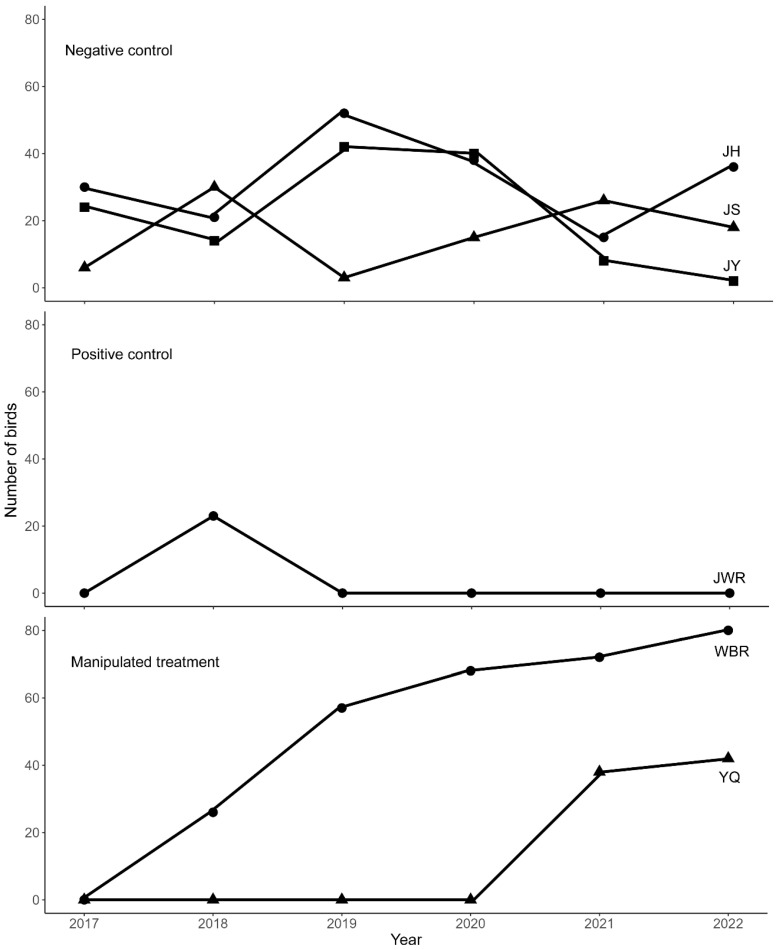
Number of blue-tailed bee-eaters (*Merops philippinus*) recorded at the six study sites during the breeding season from 2017 to 2022. JH: Jinhui; JY: Jinshawanyi; JS: Jinshawansi; JWR: Jinshawan Wetland Reserve; YQ: Yongqing; WBR: Wuyuanhe Bee-eater Reserve.

**Figure 4 animals-13-01112-f004:**
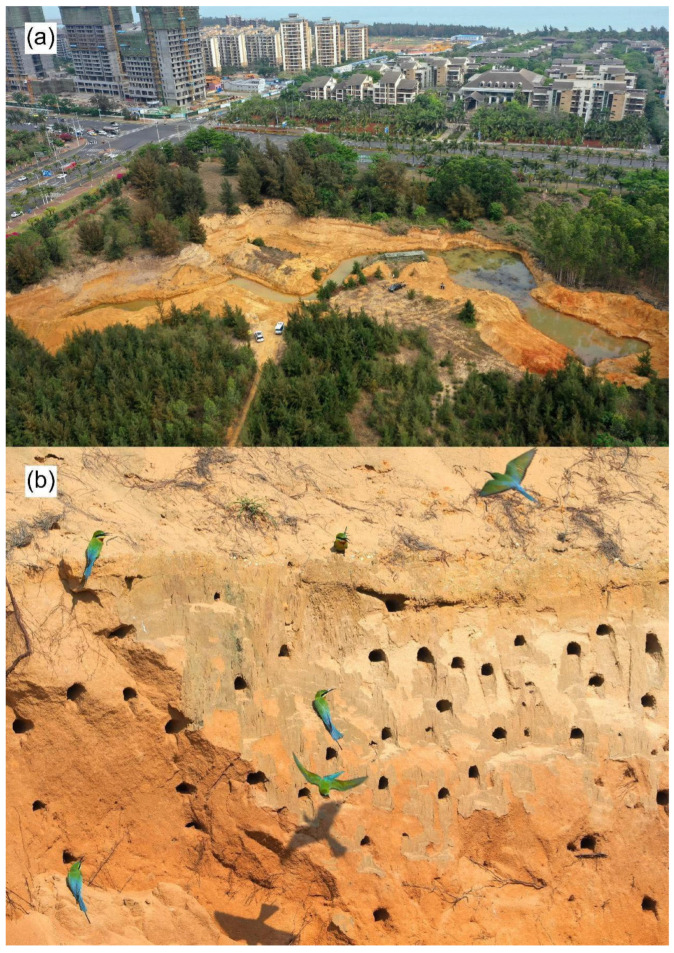
Wuyuanhe Bee-eater Reserve, where manipulated sand digging was conducted in the years 2017 to 2022, 1 month before the blue-tailed bee-eaters (*Merops philippinus*) initiated reproduction. (**a**) Aerial view of the reserve area. (**b**) A sand wall with blue-tailed bee-eaters and their nesting cavities.

**Table 1 animals-13-01112-t001:** Details of the six study sites from 2017 to 2022.

Study Site	Area (ha)	Group	Description
Jinhui (JH)	1.0	Negative control	exploitation by sand digging in 2017–2022 without manipulation
Jinshawanyi (JY)	1.5		
Jinshawansi (JS)	2.0		
Jinshawan Wetland Reserve (JWR)	4.0	Positive control	no sand digging in 2017; exploitation by sand digging occurred in 2018 and was then strictly prohibited from 2019 to 2022
Yongqing (YQ)	1.8	Manipulated treatment	no sand digging in 2017–2020; manipulated sand digging was performed in 2021 and 2022
Wuyuanhe Bee-eater Reserve (WBR)	1.2		no sand digging in 2017; manipulated sand digging was performed in 2018–2022

**Table 2 animals-13-01112-t002:** Results of the generalized linear mixed model based on Markov chain Monte Carlo techniques (MCMC-GLMM).

	Posterior Mean of Estimates	Lower 95% CI	Upper 95% CI	*P*-MCMC
Intercept	16,989.98	2839.90	31,363.82	0.026
Treatment	−13,481.94	−21,825.19	−6083.02	0.004
Year	−8.41	−15.54	−1.41	0.026
Treatment × Year	6.68	3.02	10.82	0.004

The number of birds was the response variable, while treatment refers to no sand digging, exploitative sand digging, and manipulated sand digging. CI, confidence interval.

**Table 3 animals-13-01112-t003:** Results of logistic regression with Poisson function of error distribution.

	Estimate	S.E.	*Z*	*P*
Intercept	202.40	102.58	1.97	0.049
Group	−342.03	63.87	−5.36	<0.001
Year	−0.10	0.05	−1.94	0.053
Group × Year	0.17	0.03	5.35	<0.001

The number of birds was the response variable, while the group refers to the negative control, positive control, and manipulated treatment.

## Data Availability

The data presented in this study are available on request from the corresponding author.

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
