# Peer review of "Illegal and Exploitative Sand-Digging Activities Could Be Managed to Create Suitable Nesting Habitats for Blue-Tailed Bee-Eaters (Merops philippinus)"

_animals, 2023, doi:10.3390/ani13061112_

Round 1

Reviewer 1 Report (New Reviewer)

As all of my previous comments and concerns have adequately been addressed, I am recommending the acceptance of the article.

Author Response

Thank you very much for your reivew. Your comments were really helpful for the improvement of our manuscript.

Reviewer 2 Report (New Reviewer)

The manuscript has significantly improved compared to the initial version. The statistical analyses used are adequate and explain the effect of the treatment, group and year on the number of bee-eaters.

However, these results are not adequately discussed. The authors have limited themselves to improving the English of the discussion section, but the changes must be deeper. They should incorporate into the discussion the main results obtained through statistical analysis, critically analyzing the effect of the type of treatment, group and year on the abundance of bee-eaters

Author Response

Thank you very much for your comments. They were really helpful for the improvement of our manuscript. We have added more discussion according to your suggestion. Please see lines 265-267, 270-272, 276-280, 297-303, and 327-329 in Discussion section.

Round 2

Reviewer 2 Report (New Reviewer)

Comments to the changes

Introduction

Some changes made in the Introduction have not significantly improved the text. Specifically:

-           in the context of this work, the paragraph referring to invasive species (lines 50-52) does not contribute anything relevant.

-          the comment about ecological traps (lines 53-58) is adequate for this work, but all the examples indicated below (lines 58-76) are sufficiently known and unnecessary. I suggest to delete them.

Materials and Methods

-          Lines 150-153. Text can be shortened and improved. I suggest: “We used binoculars Swarovski EL 10 × 32 WB to observe and count all individuals of bee-eaters in each study site. Because the bee-eaters become colonies around the sand walls during breeding season, it was feasible to count all individuals in the study sites.”

Results

-          Line 243. From my point of view, the inclusion of a new figure (Figure 5) does not provide new relevant information in the context of this work.

Discussion

The paragraphs introduced in the discussion have improved the content.

Author Response

Thank you very much for your comments. We have improved the manuscript according to your suggestion.

We have deleted some unnecessary content in Introduction, shortened the Methods, and removed the Figure 5.

This manuscript is a resubmission of an earlier submission. The following is a list of the peer review reports and author responses from that submission.

Round 1

Reviewer 1 Report

1. The article specify anthropogenic intervention as an alternative way of conservation strategy other than completely protection. While the content describing exploitative activities in the introduction, however, might mislead that reclaiming natural landscapes can be beneficial to wildlife. Exploitation, especially in large-scale, still cause disastrous biodiversity lost.

2. This article oversimplify the demands of blue-tailed bee-eater during breeding. It is no doubt that bare sandy or loamy cliff attract blue-tailed bee-eater to excavate burrow, but they also need different types of habitat to survive, such as foraging site with insect resource, communal nocturnal roosting site to make it through the night safely. Only create nesting site without other habitats will cause another type of ecological trap that have worse body conditions and breeding performance.

 3. Method of counting individuals appear in the colony is too rough, needed to be described more clearly. 

4. Literature review in this article is insufficient, many points of view are mentioned based on personal experience.  Those points should be able to find references easily. 

Reviewer 2 Report

This is a very interesting article that illustrates the coexistence between humans and wildlife, by slightly adjusting development activities to favor habitat use by birds. The authors are also cautious in their recommendations, highlighting that it is not a win-win situation by default, as it requires carefully planned interventions, based on scientific evidence such as summarized in this study.

Reviewer 3 Report

You describe an interesting and useful exercise in practical conservation.

 Unfortunately, from the perspective of presentation in the scientific literature as an experiment, your study suffers from a fundamental design problem. There is nothing in your results that demonstrates the “ecological trap” you report (as key in Abstract) for the negative control sites. The birds persisted at all those sites. Were all, or only some, or sometimes none of their nests destroyed by the sand mining? Did they return because some nests were successful the previous year, or because there were fresh vertical sand walls available? You don’t present any data on the availability of suitable nest surfaces, of the persistence or destruction of these surfaces through the nesting period, nor of nest success. We are left wondering why bee-eater numbers fluctuated at negative control sites in the way they did, and this is central to the point of the paper.

Your statistical analysis is invalid and inappropriate:
* Fisher’s Exact Test requires that data points be independent, but your birds nest in pairs and colonially;
* these tests do not directly, nor usefully indirectly address the key question, which is a comparison of the pro-active treatment sites with the control sites; and

* repeated pair-wise comparison of years creates a severe multiple-testing problem.

I think statistical tests are unnecessary here. Given practical limitations to replication, a good clear graph is all that is required.

However, your graphs are not clear. Fig. 5 is obscure, seemingly presented to support the (invalid) Fisher’s Exact Tests, and is best deleted. Use one or the other, not both, of the graphs in Fig. 3, and present it much more clearly, with colour coding of treatments, not sites (you can still label the lines).

 Abstract: please re-order this as you tell us the conclusion (lines 21-24) before you tell us your results (lines 24-26)

Fig. 6 and related text is irrelevant and should be deleted.

I recommend you present this in simpler, clearer form and with substantial revision of English expression as well (see below).

English expression

Please review this comprehensively. I illustrate the sort of numerous small issues I found in the Simple Summary and Abstract alone:

l12: please explain what you mean by “ a similar ecological trap”

l12: replace “for” with “to create”

l13: replace “indicated” with “indicates”, or better with “illustrates”

l16: delete “worldwide” as it is redundant; delete “Such” as it is redundant; delete “influential and” as this doesn’t contribute anything

l22: again, the phrase “a similar ecological trap” is unclear

l22, “that inflicted damage on the reproduction” – can you be more specific, e.g. “destroyed the nests”

l23: delete “planned manipulation and” as it is redundant

l23: replace “avoidance of” with “avoiding”

l25: “persistently”?

l25: delete “during the six-year experiment” as you’ve already told us that

l25-6: replace “manipulated sand digging” with “when digging during the breeding season was avoided”

l26: delete “Nevertheless,” as it is redundant

l28: delete “in this case” as it is redundant

Minor comments

l43: it isn’t clear what you mean by “results in a high degree of niche overlap”

l53: italicise “Molothrus”

l62:  “not common”. Please cite several. I’m sure they exist

l78: what do you mean by “near passerine”? Bee-eaters are non-passerines.

l95: spell “Field”

l96 et al.: section 2.2 is very poorly expressed and could be written much more clearly and concisely

l121: what is a “waterfall figure”?

l133: to what sentence does “Figure 3”refer. Citation should be included within a sentence.

l135: please do not refer to the lines within figures as Fig. 3a, 3b etc. as this confuses with multiple graphs within a figure

Fig. 5: I have no idea what this graph means